# MIMEx:
# Intrinsic Rewards from Masked Input Modeling

**Toru Lin**
University of California, Berkeley
`toru@berkeley.edu`

**Allan Jabri**
University of California, Berkeley
`ajabri@berkeley.edu`

## Abstract

Exploring in environments with high-dimensional observations is hard. One promising approach for exploration is to use intrinsic rewards, which often boils down to estimating "novelty" of states, transitions, or trajectories with deep networks. Prior works have shown that conditional prediction objectives such as masked autoencoding can be seen as stochastic estimation of pseudo-likelihood. We show how this perspective naturally leads to a unified view on existing intrinsic reward approaches: they are special cases of conditional prediction, where the estimation of novelty can be seen as pseudo-likelihood estimation with different mask distributions. From this view, we propose a general framework for deriving intrinsic rewards – **M**asked **I**nput **M**odeling for **Ex**ploration (MIMEx) – where the mask distribution can be flexibly tuned to control the difficulty of the underlying conditional prediction task. We demonstrate that MIMEx can achieve superior results when compared against competitive baselines on a suite of challenging sparse-reward visuomotor tasks.

## 1   Introduction

In supervised learning (SL), it is often assumed that data is given, i.e. the data collection process has been handled by humans. In reinforcement learning (RL), however, agents need to collect their own data and update their policy based on the collected data. The problem of *how* data is collected is known as the exploration problem. Agents face the *exploration-exploitation trade-off* to balance between maximizing its cumulative rewards (exploitation) and improving its knowledge of the environment by collecting new data (exploration). Exploration is therefore of central importance in the study of RL; it is the only way through which an agent can learn decision-making from scratch.

Simple exploration heuristics like using $\epsilon$-greedy policy [22, 23] and adding Gaussian action noise [21, 30, 31] are often too inefficient to solve challenging tasks with sparse reward. Meanwhile, designing dense reward functions is intractable to scale to many tasks across environments, due to the amount of domain knowledge and engineering effort needed. Training agents with intrinsic rewards, i.e. providing exploration bonuses that agents jointly optimize with extrinsic rewards, is a promising way to inject bias into exploration. This exploration bonus is often derived from a notion of "novelty", such that agent receives higher bonus in more "novel" states. In classical count-based methods [35], for example, the exploration bonus is derived from state visitation counts directly. To scale to high dimensional states, recent works often derive the exploration bonus from either (1) *pseudo-counts* [4, 24, 39] (i.e. approximate state visitation counts) or (2) prediction error from a proxy model. In the latter case, the proxy model is responsible for representing the relative novelty of states, for example with respect to learned forward models [25, 34] or randomly initialized networks [8].

While count-based (including pseudo-counts) methods and prediction-error-based methods are typically viewed as distinct approaches for intrinsic reward [1, 2, 20], these methods often boil down to similar formulations in practice. That is, they rely on the same underlying idea of estimating

---

Code available at `https://github.com/Toru0w0/mimex`.

37th Conference on Neural Information Processing Systems (NeurIPS 2023).

novelty by approximating likelihood, while different in the quantities they model: states, transitions, trajectories, etc. Recent works have demonstrated how conditional prediction objectives such as masked language modeling are connected to pseudo-likelihood [29, 43], one way of approximating likelihood. Under this view, approaches that estimate novelty can be viewed as modeling different conditional prediction problems, or masked prediction problems with different mask distributions.

From this perspective, we propose **M**asked **I**nput **M**odeling for **Ex**ploration (MIMEx), a generalized framework for intrinsic reward methods. On a high level, MIMEx allows for flexible trajectory-level exploration via generalized conditional prediction. To evaluate on hard-exploration problems with high-dimensional observations and dynamics, we develop a benchmark suite of eight challenging robotic manipulation tasks that involve realistic visuomotor control with sparse rewards. We show that MIMEx achieves superior results when compared with common baselines, and investigate why it works better than other approaches through extensive ablation studies.

## 2 Preliminaries

### 2.1 From RL Exploration to Intrinsic Reward

Among approaches for RL exploration, one of the most successful is to jointly optimize an *intrinsic reward* in addition to the environment reward (a.k.a. the extrinsic reward). At each time step, the agent is trained with reward $r_t = r_t^e + \beta \cdot r_t^i$, where $r_t^e$ is the environment reward, $r_t^i$ is the intrinsic reward, and $\beta$ is a parameter to control the scale of exploration bonus. The intrinsic reward approach is flexible and general, as it simply requires modifying the reward function. This also means that intrinsic reward function can be learned and computed in a manner compartmentalized from the policy, allowing one to leverage models and objectives that scale differently from policy optimization.

The key principle underlying intrinsic reward approaches is *optimism in the face of uncertainty* [7], which has been empirically observed to be effective for encouraging active exploration [37]. In other words, the intrinsic reward is supposed to encourage agents to visit states that are less frequently visited, or pursue actions that lead to maximum reduction in uncertainty about the dynamics; note that the latter is equivalent to visiting *transitions* that are less frequently visited.

### 2.2 Intrinsic Reward and Conditional Prediction

Works in intrinsic reward literature tend to draw on different concepts such as novelty, surprise, or curiosity, but their practical formulation often follows the same principle: to encourage the agent to visit novel states or state transitions, it is desirable for $r_t^i$ to be higher in less frequently visited states or state transitions than in frequently visited ones. Deriving intrinsic reward therefore requires knowledge on state density or state transition density.

Recent intrinsic reward methods can be categorized into either pseudo-counts-based or prediction-error-based [2]. In pseudo-counts-based methods [4, 8, 24], state density is modeled directly; in prediction-error-based methods [18, 25], a forward model is learned and prediction error is used as a proxy of state transition density. Under this view, we can unify existing intrinsic reward methods into one framework, where only the underlying conditional prediction problem differs.

### 2.3 Pseudo-Likelihood from Masked Prediction Error

Prior works [29, 43] have shown how the masked autoencoding objective in models like BERT [9] can be viewed as stochastic maximum pseudo-likelihood estimation on a training set of sequences of random variables $X = (x_1, ..., x_T)$. Under this objective, the model approximates the underlying joint distribution among variables, by modeling conditional distributions $x_t|X_{\setminus t}$ resulting from masking at position $t$. Despite being an approximation, the pseudo-likelihood has been shown to be a useful proxy, e.g. as a scoring function for sampling from language model [29].

More concretely, [43]* define pseudo log-likelihood as:

$$\text{PLL}(\theta; D) = \frac{1}{|D|} \sum_{X \in D} \sum_{t=1}^{|X|} \log p(x_t|X_{\setminus t}), . \tag{1}$$

where $D$ is a training set. To more efficiently optimize for the conditional probability of each variable in a sequence given all other variables, they propose to stochastically estimate under a masking

---

*While [43] was later retracted due to an error, our argument does not depend on the part with error. We provide additional clarification on the validity of our claim in Appendix A.1.

distribution $\tilde{t}_k \sim \mathcal{U}(\{1, \ldots, |X|\})$:

$$\frac{1}{|X|} \sum_{t=1}^{|X|} \log p(x_t|X_{\setminus t}) = \mathbb{E}_{t \sim \mathcal{U}} \left[ \log p(x_t|X_{\setminus t}) \right] \approx \frac{1}{K} \sum_{k=1}^{K} \log p(x_{\tilde{t}_k}|X_{\setminus \tilde{t}_k}),$$

While this is defined for categorical variables, for continuous variables we can simply consider reconstruction error on masked prediction as regressing to a unit Normal distribution of variables such as observations, features, and trajectories. From this perspective, we can view different cases of conditional prediction as different forms of maximum pseudo-likelihood estimation and implement them as masked autoencoders.

| Method | Masked | Prediction Objective |
|---|---|---|
| Pseudo-counts [4] | subset of state $s_T$ | $\|s_T - f(\text{mask}(s_T))\|^2$ with mask on state dims |
| RND [8] | current-step feature $x_T$ | $\|x_T - f(\text{mask}(x_T))\|^2$ with $x_T = \phi(s_T)$ under a random network $\phi$ |
| ICM [25] | next-step feature $x_{T+1}$ | $\|[x_{T+1}, x_T, a_T] - f([\text{mask}(x_{T+1}), x_T, a_T])\|^2$ with $x_T = \phi(s_T)$ under inverse dynamics encoder $\phi$ |

Table 1: Examples of existing intrinsic reward methods viewed as masked prediction problems with different mask distributions.

## 3 Sequence-Level Masked Autoencoders

The observations outlined in Section 2, in particular the link between masked prediction and pseudo-likelihood, allow us to draw connections between different classes of intrinsic reward methods. Prediction-error-based methods consider the problem of conditionally predicting $s_{t+1}$ from $(s_t, a_t)$ by masking $s_{t+1}$ from $(s_{t+1}, s_t, a_t)$ tuples, which can be viewed as approximating the pseudo-likelihood of the next state under a Gaussian prior. Pseudo-counts-based methods approximate the likelihood of $s_t$, which can be achieved by marginalizing the prediction error across maskings of $s_t$. As summarized in Table 1, we can view these approaches as estimating pseudo-likelihood via masked prediction with different mask distributions.

Inspired by this perspective, we propose **M**asked **I**nput **M**odeling for **Ex**ploration (MIMEx), a generalized framework for intrinsic reward methods. Simply put, MIMEx derives intrinsic reward based on masked prediction on input sequences with arbitrary length. Such a framework naturally lends itself to greater control over the difficulty of the underlying conditional prediction problem; by controlling how much information we mask, we can flexibly control the signal-to-noise ratio of the prediction problem and thus control the variance of the prediction error. Moreover, it provides the setup to fill a space in literature: trajectory-level exploration. Existing approaches framed as conditional prediction often consider one-step future prediction problems, which can saturate early as a useful exploratory signal [8, 16, 25]. While longer time-horizon prediction problems capture more complex behavior, they can suffer from high variance [27]. MIMEx allows for tempering the difficulty of prediction by varying input sequence length; by setting up conditional prediction problems on trajectories, we can obtain intrinsic rewards that consider transition dynamics across longer time horizons and extract richer exploration signals. We can also easily tune the difficulty of the prediction problem, by varying both the input length and the amount of conditioning context given a fixed input length.

The fact that MIMEx sets up a masked prediction learning problem comes with several additional benefits. First, masked autoencoding relies on less domain knowledge compared to methods like contrastive learning, and has proven success across many different input modalities [9, 10, 17, 32, 43]. Moreover, we can leverage standard architectures such as those used in masked language modeling [9] and masked image modeling [17], for which the scalability and stability have been tested. MIMEx can be added to any standard RL algorithm as a programmatically simple module that encourages exploration.

We describe the implementation details of MIMEx in the remainder of this section. An overview of our framework is shown in Figure 1.

### 3.1 Sequence Autoencoders for Exploration

Given a sequence of trajectory information stored in the form of tuples $(o_k, a_k, r_k^e)$ for $k = 1, ..., t$, we now describe how to derive intrinsic reward $r_t^i$ for time step $t$.

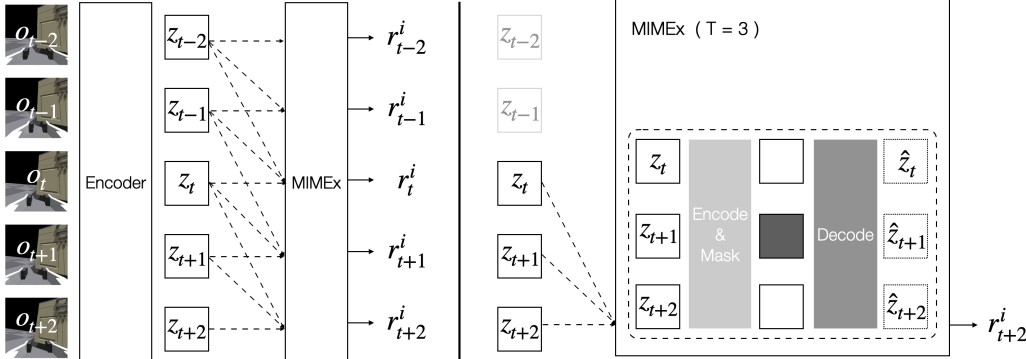

Figure 1: **Overview.** The overall schematic of MIMEx, in the case of exploration sequence length $T = 3$. (*Left*) Given a trajectory of high-dimensional observations $\{o_k\}$, the base RL encoder first encodes them into latent embeddings $\{z_k\}$; MIMEx then takes in $\{z_k\}$ and derives the corresponding intrinsic rewards $\{r_k^i\}$ for $k = t-2, t-1, ..., t+2$. (*Right*) For each timestep $k$, MIMEx performs online masked prediction on the sequence of embeddings up to $T$ steps before $k$, padding missing timesteps with zeros. The masked prediction loss $L_k$ is used as the raw intrinsic reward, i.e. $L_k = r_k^i$.

**Input sequence generation**  We define a parameter $T$ that represents the length of sequence we use to derive the exploration bonus. From the selected time step $t$, we extract the sequence of inputs up to $T$ steps before $t$, including time step $t$. When $t >= T$, we take sequence $(o_{t-T+1}, ..., o_{t-1}, o_t)$; when $t < T$, we pad observations at time steps before the initial step with zeros such that the input sequence length is always $T$. We use a Transformer-based encoder to convert each input observation $o_k$ to a latent embedding $z_k$, obtaining an input sequence $(z_{t-T+1}, ..., z_t)$.

**Online masked prediction**  Given the generated input sequences, we train a masked autoencoder model online using the sequences as prediction target. Specifically, we treat observation embedding $z_k$ at each time step $k$ as a discrete token, randomly mask out $X\%$ of the tokens in each sequence using a learned mask token, and pass the masked sequence into a Transformer-based decoder to predict the reconstructed sequence. The random mask is generated with a uniform distribution. The more "novel" an input sequence is, the higher its masked prediction error is expected to be. MIMEx is compatible with a wide range of high-dimensional inputs across different modalities, thanks to the scalability of its architecture and the stability of its domain-agnostic objective.

### 3.2 Masked Prediction Loss as Intrinsic Reward

We use the scalar prediction loss generated during the online masked prediction as an intrinsic reward, i.e. $r_t^i = L_t$. The intrinsic reward is used in the same way as described in Section 2, i.e. $r_t = r_t^e + \beta \cdot r_t^i$ where $\beta$ is a parameter to control the scale of exploration bonus. Intuitively, this encourages the agent to explore trajectory sequences that have been less frequently encountered.

### 3.3 Backbone RL Algorithm

Our exploration framework is agnostic to the backbone RL algorithm since it only requires taking a trajectory sequence as input. In Section 5, we empirically demonstrate how MIMEx can work with either on-policy or off-policy RL algorithms, using PPO [31] and DDPG [21, 33] as examples.

## 4 PixMC-Sparse Benchmark

### 4.1 Motivation

Exploration can be challenging due to sparsity of rewards, or high-dimensionality of observation and action spaces paired with complex dynamics. Existing benchmarks for exploration algorithms span a spectrum between two ways of controlling the sparsity of reward: (1) long task horizon; (2) high-dimensional observations (e.g. pixels). Either side of the spectrum can contribute to increased sample complexity, along the time dimension or the space dimension. While existing works on deep RL exploration often evaluate on popular benchmarks such as the Arcade Learning Environment [6] and DeepMind Control Suite [40], these benchmarks often involve densely engineered rewards or deep exploration in environments with simple dynamics (e.g. Montezuma's Revenge) [3, 12, 38].

Identifying a missing space of exploration benchmarks that involve higher-dimensional observation inputs and more complex dynamics, we construct a challenging task suite named PixMC-Sparse (Pixel Motor Control with Sparse Reward). The suite includes eight continuous control tasks, with realistic dynamics and egocentric camera views of a robot as observation. PixMC-Sparse is built on PixMC [44] as an extension to the original suite.

## 4.2 From PixMC to PixMC-Sparse

The original PixMC task suite does not necessarily pose challenging exploration problems due to densely shaped rewards (see Appendix A.2 for details of the original PixMC reward functions). Therefore, we modified the tasks to reflect more realistic hard-exploration challenges. To summarize, we sparsify the reward signals by removing certain continuous distance-based rewards from each original task. We show details of how each task's reward function is modified in Table 2, and include additional annotated visualization in Appendix A.3.

## 4.3 Evaluation Protocol

The difficulty level of each hard-exploration task presented in PixMC-Sparse can be tuned by (1) increasing the environment's time or space resolution, or (2) minimizing reward shaping. Either approach reduces the extent to which the benchmark is saturated, and the latter is itself a practical and realistic motivation for studying exploration in RL. In Table 3, we show a sample curriculum to minimize reward shaping for *Cabinet*, *Pick*, and *Move* tasks, starting from the fully dense reward functions (included in Appendix A.2).

| Task | Removed Dense Reward Terms |
|------|----------------------------|
| *FrankaReach* | distance to goal |
| *KukaReach* | distance to goal |
| *FrankaCabinet* | distance to handle |
| *KukaCabinet* | finger distance to handle, thumb distance to handle |
| *FrankaPick* | distance to object |
| *KukaPick* | finger distance to object, thumb distance to object |
| *FrankaMove* | distance to object |
| *KukaMove* | finger distance to object, thumb distance to object |

Table 2: We define PixMC-Sparse tasks by removing reward terms from the reward functions of PixMC tasks. In the table, we show details of how each task reward function is modified.

| Task | Example Steps to Sparsify Rewards |
|------|-----------------------------------|
| *FrankaCabinet* | 1. Remove "distance to handle" term |
|  | 2. Remove "distance to goal" term |
| *KukaCabinet* | 1. Remove "finger / thumb distance to handle" terms |
|  | 2. Remove "distance to goal" term |
| *FrankaPick* | 1. Remove "distance to object" term |
|  | 2. Remove "distance to goal" term |
| *KukaPick* | 1. Remove "finger / thumb distance to object" terms |
|  | 2. Remove "distance to goal" term |
| *FrankaMove* | 1. Remove "distance to object" term |
|  | 2. Remove "distance to goal" term |
| *KukaMove* | 1. Remove "finger / thumb distance to object" terms |
|  | 2. Remove "distance to goal" term |

Table 3: A sample curriculum to gradually increasing exploration difficulty for each task in PixMC-Sparse.

In Figure 2, we present a case study of evaluating exploration algorithms on increasingly sparse *KukaPick* tasks. Task success is defined by agent picking up an object on table using a Kuka arm.[†]

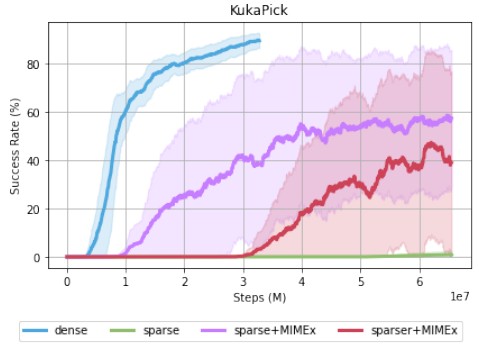

Figure 2: **Evaluating exploration algorithms with increasingly sparse reward.** We show an example on PixMC-Sparse's *KukaPick* task. When using fully dense reward, the base RL agent with random action noise exploration can already achieve success rates of over 80% consistently (*dense*). With one dense reward term removed, the base RL agent completely fails to solve the task (*sparse*). With the addition of our exploration algorithm, MIMEx, some success is observed and the success rates again start to provide useful information for evaluation (*sparse+MIMEx*). With more dense reward terms removed, performance of MIMEx agent is further away from saturation (*sparser+MIMEx*). We can continue to develop and evaluate more powerful exploration algorithms by using increasingly sparse reward.

## 5 Experiments

In this section, we show that MIMEx achieves superior results and study why it works better than other approaches. We start by evaluating MIMEx against three baseline methods, on eight

---

[†]More details on the tasks can be found in Appendix A.3.

tasks from PixMC-Sparse (Section 4) and six tasks from DeepMind Control suite [40]. Then, we present ablation studies of MIMEx on PixMC-Sparse to understand specific factors that contribute to MIMEx's performance and obtain insights for its general usage. We report all results with 95% confidence intervals and over 7 random seeds.

## 5.1 Implementation Details

We describe details of MIMEx implementation, independent of the base RL algorithms. On a high level, implementation of the MIMEx model is similar to that of MAE [17]. The model consists of an **encode & mask** module and a **decode** module.

Given an input sequence $x$ with shape $(B, L, D)$ where $B$ is the batch size, $L$ is the sequence length, and $D$ is the feature dimension, the **encode & mask** module first projects the last dimension of $x$ to the encoder embedding size through a fully-connected layer; the corresponding positional embeddings are then added to the time dimension (length $L$) of $x$. We denote this intermediate latent vector as $z$. We apply random masking with uniform distribution to the latent vector $z$, following the same protocol used in MAE [17]. The masked latent vector $z_m$ is then passed through a Transformer-based encoder.

The **decode** module first inserts mask tokens to all masked positions of $z_m$, then passes $z_m$ to a Transformer-based decoder. Finally, another fully-connected layer projects the last dimension of $z_m$ back to the input feature dimension $D$. We calculate the reconstruction loss on only the masked positions, and use the loss as intrinsic reward.

For all environments, we use a batch size of 512, the Adam [19] optimizer, and a MIMEx learning rate of 0.0001. We use a mask ratio of 70% for all tasks; a $\beta$ (exploration weight) of 0.05 for *Reach* tasks and 0.05 for *Cabinet, Pick, Move* tasks. For the encoder, we use an embedding dimension of 128, with 4 Transformer blocks and 4 heads; for the decoder, we use an embedding dimension of 64, with 1 Transformer block and 2 heads. We run each experiment on an NVIDIA A100 GPU.

## 5.2 Comparison with Baselines

We compare MIMEx to three RL exploration baselines: random action noise (*noise*), intrinsic curiosity module [25] (*icm*) and random network distillation [8] (*rnd*). We choose these baselines for their generality in formulation and robustness in performance. We show that MIMEx can be flexibly incorporated into both on-policy algorithms and off-policy algorithms: for tasks in PixMC-Sparse, we implement MVP [44] with PPO [31] (an on-policy algorithm) being the core RL algorithm; for tasks in DeepMind Control suite (DMC), we implement DrQv2 [45] with DDPG [33] (an off-policy algorithm) being the core RL algorithm. Note that MVP and DrQv2 are the respective state-of-the-art algorithm on each environment. Results are shown in Figure 3.

For PixMC-Sparse, we find that MIMEx outperforms all baselines on all tasks, in terms of both final task success rate and sample efficiency. Importantly, the performance gap between MIMEx and baseline methods becomes more pronounced on the harder *Pick* and *Move* tasks, where exploration also becomes more challenging.

For DMC, we find that MIMEx achieves higher average performance than all baselines on 2 out of 6 tasks (*cartpole_swingup_sparse*, *quadruped_walk*), while having similar average performance as the *noise* and *rnd* baselines on the other 4 tasks (*acrobot_swingup*, *finger_turn_easy*, *quadruped_run*, *finger_turn_hard*). We also observe that MIMEx exhibits lower variance compared to similarly performant baselines on all tasks. As discussed in Section 4, we find it difficult to benchmark RL exploration methods on DMC even when using high-dimensional pixel inputs, and recommend using harder exploration task suite such as PixMC-Sparse to better evaluate state-of-the-art exploration algorithms.

## 5.3 How does MIMEx improve performance?

We have conducted extensive ablation studies to make relevant design choices and understand why MIMEx works better than other approaches. For brevity, here we only present factors that result in significant improvement in task performance. Due to computational constraints, we only show results on 2 tasks from PixMC-Sparse, an easier *KukaReach* task and a harder *KukaPick* task. We summarize our findings below.

**Trajectory-level exploration**  To our knowledge, MIMEx is the first framework that successfully incorporates sequence-level intrinsic reward to solve hard exploration tasks. Intuitively, doing so

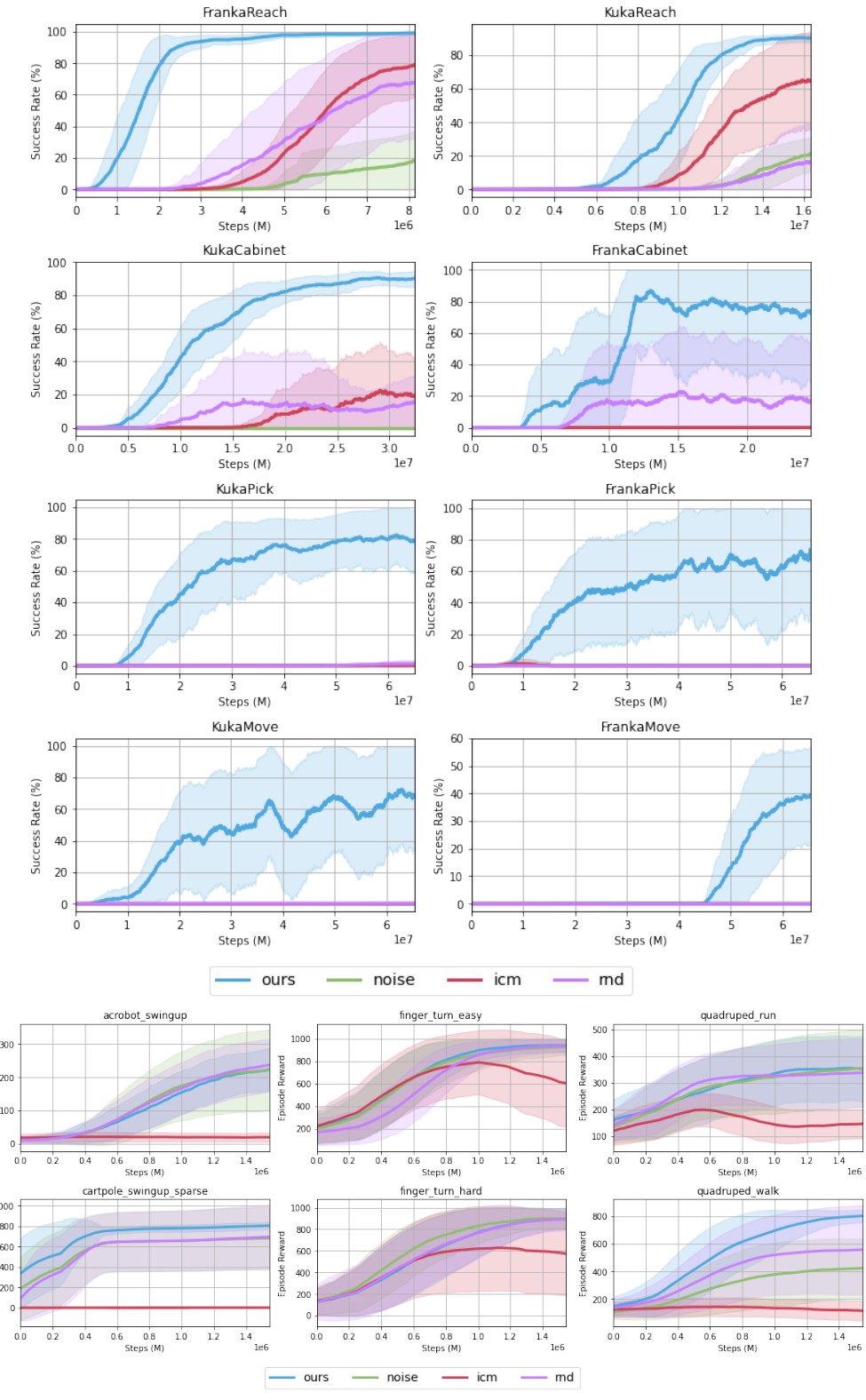

Figure 3: **Comparison with baselines.** We compare our method (*ours*) against 3 exploration baselines: random action noise (*noise*), intrinsic curiosity module (*icm*), and random network distillation (*rnd*). The top figure shows results on 8 tasks from the PixMC-Sparse suite, and the bottom figure shows results on 6 tasks from the DeepMind Control Suite.

encourages agent to search for new *trajectories* rather than one-step transitions, which may lead to more complex exploration behavior. The empirical results from varying the exploration sequence length used in MIMEx, as shown in Figure 4, suggest that using sequences with length greater than 2 (i.e. transitions that are longer than one step) indeed improves exploration efficiency.

**Variance reduction** High variance is a classical curse to RL algorithms [36]. In deriving intrinsic reward using MIMEx, a source of high variance is the random masking step during the mask-autoencode process. We hypothesize that reducing variance in conditional prediction can lead to large performance gain, and empirically test the effect of doing so. In particular, we find that a simple variance reduction technique – applying random mask multiple times and taking the average prediction error over different maskings – can greatly improve agent performance. We show our results in Figure 4.

**Model scalability** Recent advances in generative models have showcased the importance of scale, which we hypothesize also applies to the case of trajectory modeling for deriving intrinsic reward. We show our empirical results that verified this hypothesis in Figure 4. Performance gains are achieved by simply increasing the size of MIMEx's base model (specially, doubling the size of Transformer-based decoder), especially in the harder *KukaPick* environment.

## 5.4 How does varying mask distribution affect performance of MIMEx?

As discussed in Section 3, one characteristic of MIMEx is that it enables extremely flexible control over the difficulty of conditional prediction for deriving intrinsic rewards. In theory, this flexibility allows MIMEx to encompass the full spectrum of pseudo-likelihood estimation via masked prediction. We hypothesize that this would enable MIMEx to be both general and powerful: the same implementation of MIMEx can achieve superior performance compared to baselines in a wide range of settings. We are therefore particularly interested in studying how varying mask distribution affects the performance of MIMEx. Below, we show ablation studies on two attributes of MIMEx that can be easily adjusted to change the mask distribution: mask ratio and mask dimension.

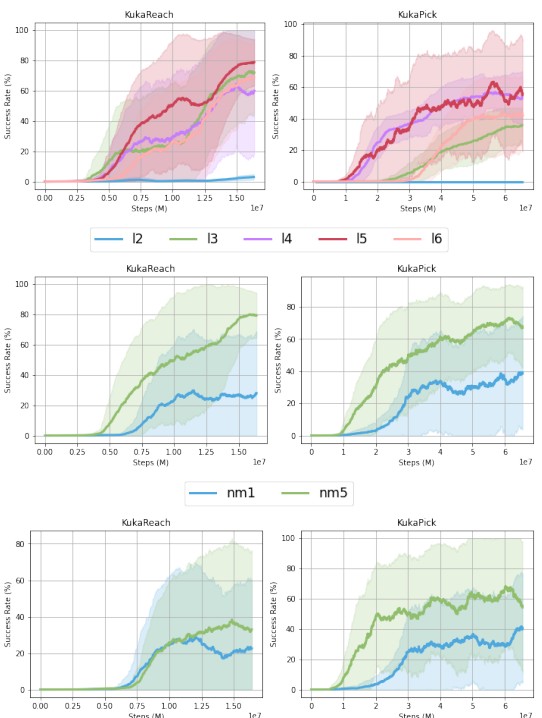

Figure 4: **(top) Varying exploration sequence length.** l*n* refers to exploration with sequence length of $n$ for $n = 2, 3, 4, 5, 6$.
**(middle) Varying number of times of masking for intrinsic reward calculation.** *nm1* is masking once, and *nm5* is masking 5 times followed by taking the average of all prediction errors.
**(bottom) Scaling MIMEx decoder.** Compared with the *smaller* model, the *larger* model uses a decoder that doubles in depth, number of output heads, and embedding dimensions.

**Mask ratio** One straightforward way to vary the mask distribution and tune the difficulty of conditional prediction is to simply adjust the mask ratio when applying random mask. Intuitively, when the mask ratio is lower, the amount of information given to the model is larger, and the conditional prediction problem becomes easier. On the two PixMC-Sparse tasks we used, we find that results from using certain fixed mask ratios exhibit high variance across tasks, for example the results from using mask ratios 20% and 50% in Figure 5. We hypothesize that, because the difficulty of exploration varies from task to task, there exists a different "optimal" mask ratio parameter for each task. In theory, MIMEx can be tuned to achieve "optimal" performance by choosing a specific mask ratio for each task. In practice, we use a fixed mask ratio of 70% for our default implementation since we find it achieve a good trade-off between hyperparameter tuning and robustness of performance.

**Mask type** Another degree of freedom of MIMEx lies in the flexibility of choosing which input dimension to apply random masks to. Fully random masking on the last dimension of input (i.e. feature dimension) will enable the most fine-grained control over the underlying conditional prediction problem. We hypothesize that this greater degree of freedom might allow MIMEx to achieve better performance on certain tasks, but might also require more substantial effort to find the optimal parameter. In Figure 5, we compare the performance of (1) masking only on the time dimension with a mask ratio of 70% and (2) masking on the feature dimension with mask ratios of 80%, 90%, and 95% on our chosen tasks. Empirically, we observe similar results between these two types of masking with the chosen mask ratios, so opted to use the more computationally efficient time-only masking in our default implementation.

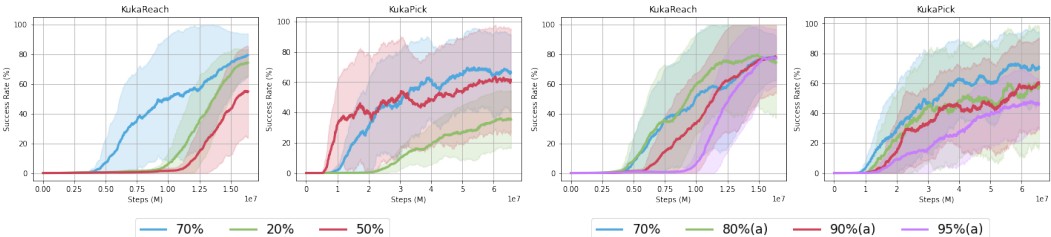

Figure 5: **(left) Varying mask ratio.** Results from masking out 20%, 50%, and 70% during the random masking step of MIMEx. **(right) Varying mask type and ratio.** Results from (1) masking only on the time dimension with a mask ratio of 70% (*70%*) and (2) masking on the feature dimension with mask ratios of 80%, 90%, and 95% (*80%(a)*, *90%(a)*, *95%(a)*).

## 6   Related Works

Undirected exploration [41] approaches add noise to individual actions (e.g. random action noise) [11], action policy (e.g. $\epsilon$-greedy), or even network weights [13]. These methods are often surprisingly strong baselines for hard-exploration problems [38].

Classical count-based methods provide another effective way to perform exploration [35]. To bring count-based exploration to high-dimensional state spaces, [4] and [24] propose pseudo-count as a useful surrogate of novelty, using CTS [5] and PixelCNN [42] as the density model respectively. [39] is another example of count-based exploration, but uses hash code instead of pseudo-count for state visitations.

[8] takes a different approach by approximating novelty with prediction error on random features. The intuition is the same – prediction error should be higher for less visited states. The closest to our work are perhaps works that generate prediction errors by learning a forward dynamics model. For example, [25, 34] can be seen as special cases of our framework in that they only perform one-step prediction. [16] is similar to our work in spirit, but learns a multi-step dynamics model with recurrent networks and generates multi-step prediction error with a BYOL-like [15] prediction objective. Relatedly, [28] derives intrinsic rewards through predicting temporally extended general value functions, though focusing on partial observable tabular environment. [18, 26] use reduction of uncertainty under the learned model as intrinsic reward; the former uses Bayesian neural network and the latter uses ensemble disagreement. In practice, these methods perform worse because it is usually hard to get a good estimate of the reduction in uncertainty with all the stochasticity inherent in modern neural networks and optimization tools.

## 7   Limitations and Future Work

Inspired by a unified perspective on intrinsic reward approaches, we present Masked Input Modeling for Exploration (MIMEx), a general framework for deriving intrinsic rewards. By using a masked autoencoding objective on variable-length input sequences, MIMEx enables flexible control over the difficulty of its underlying pseudo-likelihood estimation. By leveraging a scalable architecture and objective function, MIMEx can be easily scaled up and stabilized to solve challenging sparse-reward tasks, achieving superior performance compared to several competitive baselines.

One limitation of MIMEx that is that it can potentially hurt performance on easier tasks that do not require exploration, as in the case of exploration bonus approaches in general [38]. Moreover, while MIMEx can improve sample efficiency on hard-exploration tasks like those in PixMC-Sparse,

to tractably solve much harder problems like real-world sensorimotor learning, we might need a combination of MIMEx and stronger bias from other sources – for example expert demonstrations. Improving exploration from either angle will be important steps for future work.

While we did not investigate into how different representation learning methods affect the performance of MIMEx in this work, adding representation learning techniques is possible in the MIMEx framework and is an interesting direction for future work. Looking forward, we hope to study how MIMEx can be leveraged to transfer knowledge from pretrained representations to the exploration problem, particularly given its scalability and flexibility.

## Acknowledgements

We thank Ilija Radosavovic, Fangchen Liu, Qiyang Li, Jitendra Malik, and Alexei Efros for helpful discussions and support. TL is funded by an NSF Graduate Research Fellowship.

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

# A   Appendix

## A.1   Clarification on reference to [43]

As we make a reference to [43] – a work that was later retracted due to an error – we would like to clarify that we are aware of the error and our claims remain valid despite the error in cited work.

In [43], the authors mistakenly viewed BERT[9] as a Markov Random Field (MRF). While the original goal of [43] was to derive a procedure for sampling from masked language models (MLMs) by viewing them as MRFs, the work also inspired the use of MLM prediction error as a way for scoring sequences when decoding from a language model. The way we propose to use MLMs is similar to the latter, i.e. as a proxy metric for *scoring* sequence-level predictions of trajectory information. In other words, we do not formally treat the MLM in MIMEx as an MRF, and we do not attempt to obtain conditional distributions from which one generates samples. We also note that a correct energy-based view was later proposed in [14], which does not change the argument that we put forth either. While sampling from an energy-based model is expensive, we only seek to obtain a useful stochastic estimate of the energy function for the purpose of scoring.

## A.2   Detailed reward design of the original PixMC tasks

We present the detailed reward terms of each task in the original PixMC. The total environment reward of each task is the sum of all reward terms, each multiplied with a tunable scale parameter. "Bonus" type of reward terms are discrete while other types of reward terms are continuous.

**FrankaReach:** distance to goal (from parallel gripper); goal bonus when distance to goal is smaller than a threshold value; action penalty.

**KukaReach:** distance to goal (from humanoid hand); goal bonus when distance to goal is smaller than a threshold value; action penalty.

**FrankaCabinet:** distance to handle (of cabinet); handle bonus when mesh of parallel gripper intersects mesh of handle; open bonus when cabinet is open; distance to goal (from parallel gripper); open pose bonus when parallel gripper is within a certain pose distribution; goal bonus when distance to goal is smaller than a threshold value; action penalty.

**KukaCabinet:** distance to handle (of cabinet); handle bonus when mesh of humanoid hand intersects mesh of handle; open bonus when cabinet is open; distance to goal (from humanoid hand); open pose bonus when humanoid hand is within a certain pose distribution; goal bonus when distance to goal is smaller than a threshold value; action penalty.

**FrankaPick:** distance to object (from parallel gripper); lift bonus when object is lifted above the table; distance to goal (from parallel gripper); goal bonus when distance to goal is smaller than a threshold value; action penalty.

**KukaPick:** distance to object (from humanoid hand); lift bonus when object is lifted above the table; distance to goal (from humanoid hand); goal bonus when distance to goal is smaller than a threshold value; action penalty.

**FrankaMove:** distance to object (from parallel gripper); lift bonus when object is lifted above the table; distance to goal (from parallel gripper); goal bonus when distance to goal is smaller than a threshold value; action penalty.

**KukaMove:** distance to object (from humanoid hand); lift bonus when object is lifted above the table; distance to goal (from humanoid hand); goal bonus when distance to goal is smaller than a threshold value; action penalty.

Due to the dense reward terms, these tasks are not sufficiently challenging for benchmarking state-of-the-art exploration algorithms.

## A.3   Detailed comparison between PixMC and PixMC-Sparse tasks

We visualize partial trajectory of all PixMC-Sparse tasks in Figure 6. Below, we provide more details on how agents are rewarded differently in PixMC-Sparse compared to in PixMC.

For **Reach** tasks, agent needs to move its end effector (parallel gripper in the case of Franka; humanoid hand in the case of Kuka) such that the end effector reaches a specific goal location. In PixMC, agent receives a variable-valued reward at every time step; the reward value is continuous and inversely

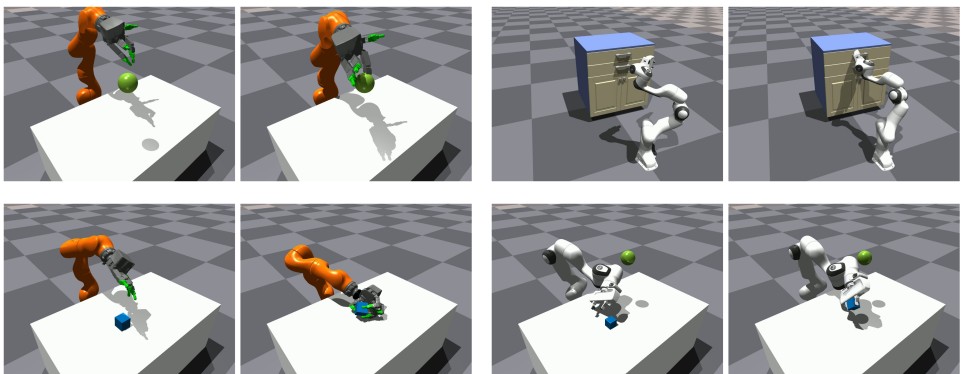

Figure 6: **Visualization of tasks.** (*Top-left*) KukaReach. (*Top-right*) FrankaCabinet. (*Bottom-left*) KukaPick. (*Bottom-right*) FrankaMove.

proportional to end effector's distance to goal location. In PixMC-Sparse, agent only receives a fixed-discrete-valued reward when its end effector intersects with a small spherical space (visualized as the green sphere) around the goal location.

For **Cabinet** tasks, agent needs to grasp the cabinet handle using its end effector, open the drawer, and keep pulling until it reaches a goal location. In PixMC, agent constantly receives a variable-valued reward inversely proportional to end effector's distance to cabinet handle. In PixMC-Sparse, agent only receives a fixed-valued reward when the mesh of its end effector intersects with the mesh of cabinet handle (visualized as the green sphere) around the goal location.

For **Pick** tasks, agent needs to grasp a cube-shaped object using its end effector and lift up the object until it reaches a goal height. In PixMC, agent constantly receives a variable-valued reward inversely proportional to end effector's distance to object, and a lift bonus when it starts to successfully lift up the object (i.e. when object is above the table). In PixMC-Sparse, agent does not receive any distance-to-object-based reward, i.e. no reward until it starts to lift up the object.

For **Move** tasks, agent needs to grasp a cube-shaped object using its end effector and lift up the object until it reaches a goal location. In PixMC, agent constantly receives a variable-valued reward inversely proportional to end effector's distance to object, and a lift bonus when it starts to successfully lift up the object (i.e. when object is above the table). In PixMC-Sparse, agent does not receive any distance-to-object-based reward, i.e. no reward until it starts to lift up the object.

### A.4 Wall-Clock Time and GPU Memory Usage Comparisons

We compare the training wall-clock time for MIMEx (*ours*) against that of 3 exploration baselines: random action noise (*noise*), intrinsic curiosity module (*icm*), and random network distillation (*rnd*). We include 3 variations of MIMEx for comparison: (1) standard model size and masking once (*smaller, nm1* in Figure 4), (2) standard model size and masking 5 times (*smaller, nm5* in Figure 4), (3) larger model size and masking once (*smaller, nm1* in Figure 4). We find that MIMEx not only achieves superior sample efficiency and performance, but also adds little wall-clock time overhead compared to the baselines.

| Method | *ours(1)* | *ours(2)* | *ours(3)* | *rnd* | *icm* | *noise* |
|---|---|---|---|---|---|---|
| Hours / 10M steps | 6.018 | 6.990 | 6.822 | 6.724 | 6.390 | 5.9 |

Similarly, we compare the average GPU memory usage for MIMEx against the baselines, using the same 3 variations of MIMEx for comparison as above. We find that while MIMEx does consume more memory resources during training, the increased memory usage is less than 36% even when comparing the most compute-intensive MIMEx version (*ours(2)*) against the least compute-intensive baseline (*noise*).

| Method | *ours(1)* | *ours(2)* | *ours(3)* | *rnd* | *icm* | *noise* |
|---|---|---|---|---|---|---|
| GPU Mem Usage / M | 18034 | 24218 | 18402 | 17930 | 17954 | 17900 |

Each number reported above was averaged from 10 experiment runs on the *KukaReach* task in PixMC-Sparse; each run was trained on a single NVIDIA A100 GPU.

## A.5 Additional Results on More Diverse Tasks and Environments

To check if MIMEx generalizes to other domains, we ran additional experiments to compare MIMEx with ICM and RND on the ALE [6] *PRIVATE EYE* and *VENTURE* environments. Below, we report all results with 95% confidence intervals, over 5 random seeds on each method (ICM, RND, MIMEx).

| $\bar{R}$ per step | 10M | 25M | 50M | 100M | 200M |
|---|---|---|---|---|---|
| RND | -164.8± 279.0 | -188.4 ± 281.9 | -160 ± 413.9 | 4105.8 ± 112.0 | 6036.4 ±7010.1 |
| ICM | -31.4 ±103.7 | 2334.8 ±2095.9 | 7864.6 ±4605.5 | 11963.8 ± 746.1 | 23377.6 ± 9720.1 |
| MIMEx | -529 ± 416.4 | 104.6 ± 658.8 | 2344.8 ±2776.8 | 7859.6 ± 6866.5 | 7717.2 ± 6239.4 |

Table 4: Mean episodic return ($\bar{R}$) at 10, 25, 50, 100, and 200 million steps of training on the *Private Eye* task.

| $\bar{R}$ per step | 10M | 25M | 50M | 100M |
|---|---|---|---|---|
| RND | 580 ± 227 | 1080 ± 144 | 980 ± 73 | 1660 ± 159 |
| ICM | 380 ± 243 | 1240 ± 202 | 1400 ± 310 | 1560 ± 133 |
| MIMEx | 460 ± 320 | 1080 ± 114 | 1040 ± 100 | 1660 ± 48 |

Table 5: Mean episodic return ($\bar{R}$) at 10, 25, 50, and 100 million steps of training on the *Venture* task.

Our results for ICM and RND are similar to those reported in prior works. For MIMEx, we only tuned the exploration beta parameter; without extensive hyperparameter tuning, MIMEx still performs comparably with ICM/RND.

## A.6 Additional Results on Varying Mask Distribution

We perform additional ablation studies to better understand the importance of mask distribution in MIMEx formulation, especially on PixMC-Sparse tasks where existing methods (ICM, RND) fail. We interpolate between the MIMEx masking distribution and a masking distribution closest to ICM/RND on the *KukaReach* task.

The following exploration sequence length and mask distributions are used: (1) 70% uniformly random mask with sequence length 5 (*l5_uniform_70*); , (2) 70% fixed mask, sequence length 5 (*l5_fixed_70*), (3) 50% fixed mask, sequence length 5 (*l5_fixed_50*), (4) 50% fixed mask, sequence length 2 (*l2_fixed_50*). For "fixed mask", we only always mask out the last X positions of the input sequence based on the mask ratio, effectively making the reconstruction problem a future-state prediction problem. All results are reported with 95% confidence intervals and over 7 random seeds.

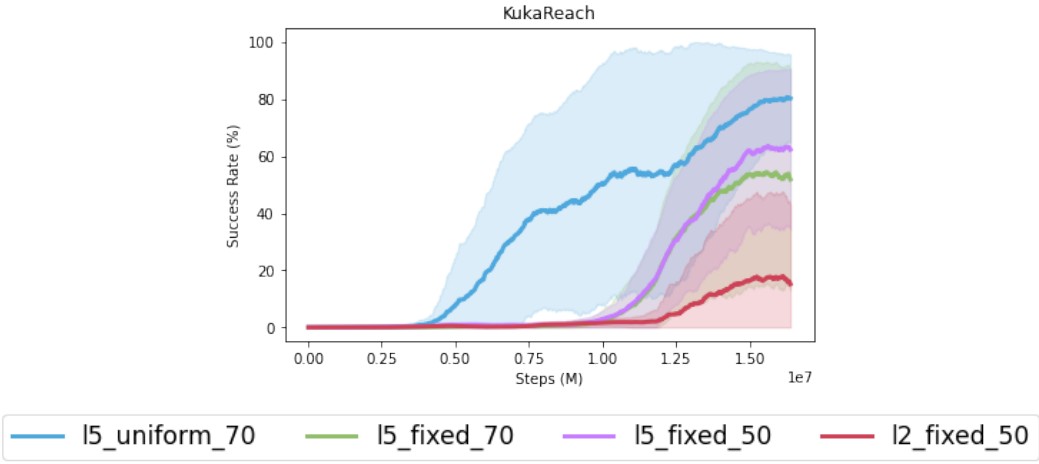

Figure 7: **Interpolating between the MIMEx masking distribution and ICM/RND masking distribution.** Performance drops when going from uniformly random mask to fixed mask when keeping exploration sequence length the same.

