# OpenReview forum: "MIMEx: Intrinsic Rewards from Masked Input Modeling"
_NeurIPS.cc/2023/Conference — NeurIPS 2023 poster_

### Official Review · Reviewer_SVND · 2023-06-27

**Soundness:** 4 excellent
**Presentation:** 4 excellent
**Contribution:** 4 excellent
**Rating:** 8
**Confidence:** 4

**Summary:**

The paper introduces a novel method for Exploration in RL, called MIMEx (Masked Input Modeling for Exploration).

Previous approaches for exploration investigated intrinsic rewards, usually computed as a measure of a state or transition’s “novelty”, adding them to extrinsic rewards (the actual task’s rewards) to enhance the algorithm’s exploration and thus performance, especially for tasks with sparse rewards.

The present paper argues that those previous intrinsic reward methods can be viewed under a unifying lens as approaches that use pseudo-likelihood estimation to estimate novelty. A general algorithm is introduced called MIMEx that computes pseudo-likelihood over entire trajectories, as the prediction loss of a masked sequence autoencoder. This loss corresponds to the intrinsic reward for an entire trajectory, and can be simply added to task rewards when running any RL algorithm.

The paper then demonstrates the effectiveness of the approach with evaluations, baseline comparisons and ablation studies on a “PixMC-Sparse” benchmark suite.

**Strengths:**

The paper is very clearly and naturally presented, with great care put into arguing for a coherent story. The work is original, arguing for a straightforward but interesting generalization of existing concepts in the RL literature.

The natural blending of theoretical justification for the work (pseudo-likelihood as novelty), application of recently popular techniques (masked sequence modeling) and experimental results make the paper highly significant. The generalization of the concept of intrinsic rewards to entire trajectories is also extremely interesting.

The experimental results are thorough, including evaluation on different benchmark settings, comparisons with other approaches, and extensive ablation studies. In particular, the ablation study showing how scaling of the autoencoder transformer affects results seems very interesting, and should warrant further investigation.

**Weaknesses:**

It seems that adding a whole masked autoencoder to compute intrinsic rewards may incur a hefty computational cost. What is the overall runtime / resource utilization of MIMEx compared to other baselines? Even if MIMEx is more computationally demanding, it can still be a good choice due to sample efficiency, especially for offline RL tasks. In any case, the paper would greatly benefit from such an analysis.

**Questions:**

A few points:
* **[a]** Can the authors address the point raised in “Weaknesses”, related to computational cost?
* **[b]** In Section 5.3, “Trajectory-level exploration”, specifically in Figure 4: Why does a sequence length of 6 reduce performance? From the authors’ presentation, it may seem that increasing trajectory length should in principle always guarantee a benefit. Why is there instead an optimum at lengths shorter than 6?
* **[c]** Minor point: For all plot figures, font sizes are too small.

**Limitations:**

I believe the authors appropriately addressed the limitations of their work.

---

> ### Author Rebuttal · Authors · 2023-08-09
>
> Thank you for the very positive feedback. We appreciate your evaluation of our work and address the questions below.
>
> **”What is the overall runtime / resource utilization of MIMEx compared to other baselines?”** (also **Questions [a]**)
>
> Thank you for the suggestion. We have included tables for both the wall-clock time and GPU memory utilization of MIMEx compared to other baselines in the global PDF, and will add this to our revised paper. While MIMEx could be more computationally demanding, the overhead in both wall-clock time and GPU memory usage is comparable to those of the baselines, so we also agree that its benefit in sample efficiency could outweigh the computational challenge.
>
>
> **Questions**
>
> **[b]** This is an interesting observation that we noticed too. We hypothesize that there is always a “sweet spot” that achieves the optimal balance of this tradeoff, rather than the longer sequence length the better: while longer exploration sequences may encourage more complex exploratory behaviors and tackle harder exploration problems, using longer exploration sequences in MIMEx may also increase the variance of prediction task and worsen performance. Our results in Figure 4, where MIMEx with exploration sequence length 6 underperforms MIMEx with shorter exploration sequence length, empirically confirm this hypothesis.
>
>
> **[c]** Thank you for the suggestion. We will increase the font sizes of all figures in our revised manuscript.

---

> > ### Comment · Reviewer_SVND · 2023-08-10
> > **Reply to Authors**
> >
> > Thank you for your rebuttal. The wall clock time and memory usages are indeed impressive.
> > I don't have any additional comments, and maintain my score as previously stated.

---

> > > ### Author Response · Authors · 2023-08-11
> > >
> > > Thank you so much again for the comments.

---

### Official Review · Reviewer_if59 · 2023-07-04

**Soundness:** 2 fair
**Presentation:** 3 good
**Contribution:** 2 fair
**Rating:** 6
**Confidence:** 4

**Summary:**

This paper proposes to use masked autoencoding (similar to MAE) in RL and use the loss as an intrinsic reward for exploration in sparse reward domains. Their method, MIMEx, does masked reconstruction of latent observation o_t, based on the previous T observations, and assigns the loss as intrinsic reward r_t. They propose that count-based or prediction error-based intrinsic rewards can be viewed in the same masked autoencoding framework, as specific inputs and masks (such as ICM and RND). They show experiments in DM Control Suite and a harder version of PixMC denoted as PixMC-Sparse, which sparsifies the shaped reward. They show that MIMEx outperforms ICM and RND baselines.

**Strengths:**

Experiment results suggest it is stronger than ICM and RND, and the method itself is relatively straightforward. The paper does clearly demonstrate evidence that this is a viable method for exploration.

**Weaknesses:**

The formulation of masked autoencoding and trying to express other methods like RND and ICM in terms of it is not a very strong argument. One of the key parts of RND is the usage of a fixed random network, which is not properly captured by a varying mask; it is captured by allowing an arbitrary transformation of the inputs. For ICM, the representation learning part from inverse control is also not captured by the formulation, but also delegated to a transformation of the inputs. Representation learning is often a key part of an exploration algorithm, which MIMEx does not directly capture. By allowing an arbitrary transformation of the inputs, the actual masking portion of MIMEx becomes less important, since masking can equally be delegated as a different transformation of the inputs, resulting in only needing a trivial mask for MIMEx. I think there should be less emphasis on this part for the paper, as the experiments also do not investigate this idea further. An example of further investigation would be to try to emulate RND/ICM or other exploration approaches through specific masks (not just random masks) and showing whether specific masks may be better or worse than uniform random masking.

I believe your reference [14] Byol-explore, does incorporate sequence-level information in its prediction error for intrinsic reward, which is not mentioned in related work.

---- After Author Rebuttals ----

After reading other reviews and the author rebuttals, I think the authors have addressed the addressable parts of my concerns, as well as have stated that they will clarify their main claim to be more nuanced. Thus I'm inclined to slightly raise my score.

**Questions:**

MIMEx seems to be encoding and decoding latents (as opposed to pixels), so are you allowing gradients from MIMEx to also flow back through to those latents? I.e. are you stopping gradients from the input/target sides? If not stopping gradients, then is there a danger that the representation will collapse to a constant? If yes, then are you relying on the RL to train the latent representations?

In your Figure 1, MIMEx only receives latent observations as input/target. Why are actions not included? Many forward model predictions rely on state and action information for prediction, including ICM. What was the reasoning behind this? If actions were included, then MIMEx would be able to more closely emulate many multi-step prediction methods such as [14] Byol-explore or SPR (Data-Efficient Reinforcement Learning with Self-Predictive Representations https://arxiv.org/abs/2007.05929) by picking a mask the only masks out observations but keeps around actions.

**Limitations:**

This paper addresses some of the limitations, but another potential limitation for this method is partial observability, especially for long horizon problems. Only using the previous T observations as input might not be a rich enough context for predicting the next observation, so there would need to be another way of adding more past information.

Another potential limitation is stochasticity. Since MIMEx is predicting observations, it will suffer from high loss if the observations are stochastic, similar to many other prediction-error-based approaches that try to directly predict the observation. RND avoids the issue by same-step prediction, while ICM relies on the representation learning of inverse control to filter our dynamics-irrelevant noise. With MIMEx, it seems like even adding white noise to observations could end up with high losses everywhere.

---

> ### Author Rebuttal · Authors · 2023-08-09
>
> Thank you for the feedback.
>
> **Strengthening the masked autoencoding argument**
>
> Thank you for the suggestion. We agree that our method only generalizes the conceptual formulation of RND or ICM as summarized in Table 1, and does not capture differences in representation learning. We will clarify this in the revised paper. To further highlight the importance of having a flexible mask distribution, we ran additional experiments where we emulate RND/ICM through specific masks.
>
> Specifically, we interpolated between the MIMEx masking distribution and a masking distribution closest to ICM/RND. Experiments with the following exploration sequence length and mask distributions are added:
> - l5_uniform_70 - seq length 5, uniformly random mask, 70% [MIMEx]
> - l5_fixed_70 - seq length 5, fixed mask, 70%
> - l5_fixed_50 - seq length 5, fixed mask, 50%
> - l2_fixed_50 - seq length 2, fixed mask, 50%
> (For “fixed mask”, we only always mask out the last X positions of the input sequence based on the mask ratio, effectively making the reconstruction problem a future-state prediction problem.)
>
>
> In the global PDF, we present results on the *KukaReach* environment, each curve with 3 seeds. We observe that our method performs better than the masking strategy corresponding to RND/ICM, indicating that the flexible mask distribution enabled by our method is beneficial:
> 1. Performance drops when going from uniformly random mask to fixed mask when keeping exploration sequence length the same.
> 2. Performance drops further when reducing the length of exploration sequence from 5 to 2. This is consistent with our ablation results presented in Fig 4.
> 3. The MIMEx variant with mask distribution closest to existing one-step exploration approaches, “l2_fixed_50”, performs the worst out of all the settings evaluated, which confirms our hypothesis that the additional flexibility in tuning masking distribution provided by MIMEx framework can positively contribute to its performance.
>
> We believe that adding representation learning techniques such as inverse models from ICM is possible in our framework and is an interesting direction for future work. We will include the additional results and update the writing in our revised paper.
>
> **Reference to BYOL-Explore**
>
> BYOL-Explore indeed incorporates sequence-level information in its prediction error, though being very different from our work in both how the information is processed (by using a recurrent neural network) and used (to build the agent’s internal representation used for future prediction). We will include BYOL-Explore in the related work section of our revised paper and clarify these differences.
>
> **Questions**
>
> Regarding how pixel observations are encoded into latents, we follow the implementation of each respective baseline (MVP [41] for PixMC-Sparse and DrQv2 [42] for DMC). Specifically, for PixMC-Sparse, the pixels are encoded into latents via a frozen pretrained ViT model into latents; for DMC, the pixels are encoded into latents via a convolution network trained online with random shift data augmentation. In the former case, the representation does not collapse since a frozen encoder model is used; in the latter case, RL is used to train the latent representations and the random shift augmentation is used to prevent collapse. Empirically, each representation learning method has been state-of-the-art in its corresponding environment.
>
> Regarding including action in the prediction objective, we chose to mask only the latent observations because doing so would make the model more general and easier to tune (as the data modality being masked is kept consistent). We also reasoned that since we mask a history of observations, it is possible that adding action information results in more redundancy as some action information can be implicitly inferred. However, we think adding action is an interesting idea and will look into this design choice in our follow-up work.
>
> **Limitations**
>
> *partial observability*
>
> Thank you for your suggestion. We agree that partial observability could be an important challenge to RL exploration methods, and that providing more information to the agent will help alleviate the challenge. Transformer-based models have been used to handle longer context length than recurrent models and to address partial observability (e.g. Chen’21, Reed’22, Brohan’22); one strength of MIMEx is therefore that it could be flexibly extended to include more data modalities or handle longer context length. Recent works have reported positive results where masked autoencoding on more general trajectory data is used for representation learning (e.g. Radosavovic’23); this is a promising sign that MIMEx could benefit from richer data too and we are excited to pursue this direction in our follow-up work.
>
> Brohan, Anthony, et al. "Rt-1: Robotics transformer for real-world control at scale." arXiv preprint arXiv:2212.06817 (2022).
>
> Chen, Lili, et al. "Decision transformer: Reinforcement learning via sequence modeling." Advances in neural information processing systems 34 (2021): 15084-15097.
>
> Radosavovic, Ilija, et al. "Robot Learning with Sensorimotor Pre-training." arXiv preprint arXiv:2306.10007 (2023).
>
> Reed, Scott, et al. "A generalist agent." arXiv preprint arXiv:2205.06175 (2022).
>
> *stochasticity*
>
> Indeed, for now we are tackling this problem by averaging multiple masks, and for PixMC-Sparse implementation we to some extent delegate this representation learning problem to the frozen pretrained ViT encoder. We do not focus on representation learning in the scope of this work, but will update our writing in Section 7 to include a discussion of this important problem in our revised paper.

---

> > ### Comment · Reviewer_if59 · 2023-08-10
> > **Masking Distributions and Actions**
> >
> > Thanks for the response to my questions! I appreciate the response and additional ablation on the masking distribution, which does give more insight into its importance.
> >
> > However I do want to emphasize my point that without actions, MIMEx is unable to properly capture many intrinsic reward methods. The main reason I am re-emphasizing this point is because this paper is making a very strong claim for the expressiveness of the MIMEx framework, which is only partially true. For example in Section 3, line 99: "Inspired by this perspective, we propose Masked Input Modeling for Exploration (MIMEx), a unifying framework for intrinsic reward methods...". The term "unifying" is used multiple times throughout the paper and abstract. MIMEx generalizes 1-step to multi-step with very flexible mask distributions, which is its strength. However without actions, MIMEx fails to model any 1-step dynamics model type of intrinsic reward, including ICM. In your ablation of [l2_fixed_50 - seq length 2, fixed mask, 50%], if you are trying to do 1-step next state prediction, having the action vs. not having the action are fundamentally different problems, and MIMEx can only model one version. Thus I think this claim of MIMEx being a "unifying" framework is too strong, and it would be better to make a more nuanced claim such as "generalized framework for state-based multi-step intrinsic reward".

---

> > > ### Author Response · Authors · 2023-08-11
> > >
> > > Thank you so much for elaborating on your point. We now understand your concern on the claim of MIMEx’s expressiveness and agree. Indeed, while MIMEx can be naturally extended to include action (by masking on a history of not only observations but also actions), we did not investigate this idea explicitly within the scope of this submission. We will tone down the writing in our next revision to make the claim more nuanced, in particular regarding the word choice of “unifying” (e.g. replace it with words like “generalized”).

---

> > > > ### Comment · Reviewer_if59 · 2023-08-14
> > > >
> > > > Thank you for addressing this. I'm inclined to slightly increase my score.

---

### Official Review · Reviewer_mFvm · 2023-07-05

**Soundness:** 2 fair
**Presentation:** 3 good
**Contribution:** 2 fair
**Rating:** 6
**Confidence:** 4

**Summary:**

The paper introduces a novel approach to exploration in reinforcement learning (RL) called Masked Input Modeling for Exploration (MIMEx). MIMEx uses a masked autoencoding objective on variable-length input sequences to derive intrinsic rewards for exploration. The paper claims that MIMEx improves exploration efficiency in sparse-reward tasks and show that MIMEx outperforms claimed competitive baselines on tasks from the PixMC-Sparse suite and the DeepMind Control Suite except for in a few cases, such as certain cases in which the reward is not sparse.

**Strengths:**

**Originality**
The paper introduces a fresh approach to exploration in reinforcement learning, applying existing concepts in a new way. This application of masked autoencoders and pseudo-likelihood estimation to the problem of exploration in sparse reward environments is at least innovative.

**Quality**
The paper is well-structured and provides a comprehensive examination of the proposed MIMEx method. The paper creates a new benchmark dataset from an existing one with non-trivial modifications, and explains these. The paper has conducted extensive experiments and provided a detailed ablation study, which adds to the quality of the paper. The paper does an adequate job at discussing it's limitations and potential future directions for its work.

**Clarity**
The paper is very well-written and mostly clear in its presentation. The paper has done an excellent job of explaining the MIMEx method, its implementation details, and the experimental setup. The explanation of dense rewards for the PixMC environment, the construction of the PixMC-Sparse, and the use of figures and tables to illustrate the results is superb.

**Significance**
MIMEx shows superior performance to certain benchmark exploration algorithms, __random action noise__, __intrinsic curiosity module__, and __random network distillation__, some of which have been shown to perform well at the time of their publication on long-horizon sparse reward tasks, such as Montezuma's Revenge. The model allows for easy adjustment of the time horizons it considers, though tuning this hyperparameter may be difficult. MIMEx is agnostic to standard RL algorithm and model choice, allowing it to be used on several problems.

---
Overall, this paper is easy to read and combines ideas into a novel exploration method that could be useful. However, it's unclear to me the generalizability of most of the results and the increased compute, memory, and wall-clock expenses wrt the horizon needed for the MIMEx module.

Thank you for providing code with your submission for both sets of benchmarks!

**I have changed my score from a 5 to a 6 as a result of the authors rebuttal, in which they show evidence of additional work that addresses the key concern I had as well as additional results clarify ambiguities I had. Thank you, Authors. I'm happy that my rebuttal helped you improve your work and that you considered and addressed my concerns by performing additional work to improve your work and sharing these results clearly in your rebuttal.**

**Weaknesses:**

The paper's main idea, while innovative in its application, is built upon existing concepts in the field, such as masked autocoders and prediction error based exploration. It uses standard latent embeddings of observations and then reconstruction of masked embeddings to compute the prediction error, which is work that has been done for years, a recent example of which is [1], work that the paper transparently references (this transparency is a strength in clarity and overall). The paper could have better highlighted the unique aspects of their approach and how it diverges from or improves upon existing methods in reinforcement learning exploration. Though PixMC-Sparse gets a discrete bonus in some environments for reaching a certain intermediate state, the paper could've evaluated MIMEx on easily accessible or created environments by easily modifying existing ones that focus on more diverse types of sparse rewards, such as discrete instead of continuous (wrt to goal distance) rewards.

- The paper's quality could be improved with a more diverse set of experimental tasks and environments. The current selection of environments for which hyperparameter and ablation studies were done do not generalize to much outside of their specific domains, so they don't demonstrate the generalizability of the MIMEx method. However, I do understand that these experiments can be computationally expensive, but I would recommend performing additional ablations on environments that differ significantly from Kuka_____ in order to maximize confidence in the generalization of the results of these ablations.
- In lines 118-119, the paper states that "MIMEx can be added...as a lightweight module..." but there is no mention, let alone evaluation wrt horizon $T$ of the increased memory footprint or wall-clock time needed to calculate the intrinsic reward.
- Later exploration methods [2, 3] that are catered toward sparsity or perform better than the baselines presented are not used as baselines. But, to the paper's defense, [3] was published just Sep 2022. However, the issue is that the most recent baseline used is RND, which was published in 2018, and there are many better ones that exist to evaluate MIMEx against.

- The paper could better justify the usage of each of the exploration baselines used for readers that may not be familiar with RND's performance on Montezuma's revenge though it does do this somewhat in the Related Works section.

- Again, while the paper shows promising results on specific tasks, its impact could be limited if the method's effectiveness doesn't extend to a broader range of tasks and environments. The paper could increase its relevance and significance by demonstrating MIMEx's effectiveness in more diverse scenarios and discussing potential applications beyond the current scope.

[1] Xiao, T., Radosavovic, I., Darrell, T., & Malik, J. (2022). Masked visual pre-training for motor control. arXiv preprint arXiv:2203.06173.
[2] Zhang, T., Rashidinejad, P., Jiao, J., Tian, Y., Gonzalez, J. E., & Russell, S. (2021). Made: Exploration via maximizing deviation from explored regions. Advances in Neural Information Processing Systems, 34, 9663-9680.
[3] Eberhard, O., Hollenstein, J., Pinneri, C., & Martius, G. (2022, September). Pink noise is all you need: Colored noise exploration in deep reinforcement learning. In The Eleventh International Conference on Learning Representations.

**Questions:**

1. Should the equality in the display mode equation in line 84 be an approximation since $|X|$ is not necessarily infinite?
2. Referring to lines 204-206, what are the internals of the transformer blocks? Could you clarify because not all transformer blocks are in the same order or contain the same properties as [4].
3. Why choose the three chosen exploration baselines, random action noise, intrinsic curiosity module, and random network distillation?
4. I'm confused by the inclusion of the sample curriculum in Table 3. Is this only an informative point, or is this something you evaluated?
5. In lines 233-234, the paper states "Trajectory-level exploration To our knowledge, MIMEx is the first framework that successfully incorporates sequence-level intrinsic reward to solve hard exploration tasks." Doesn't [5] do this implicitly as the discount factor in [5] is varied? Again, this is a relatively recent publication.
6. Minor suggestion, to make the results of Figure 2 more tangible and to improve readibility, I suggest providing a visual and explanation of the KukaPick task alongside Figure 2, even though it is in the Appendix.
7. Do you have ideas on extensions of Online masked prediction to offline RL and any advantages or disadvantages using MIMEx with this paradigm?

[4] Vaswani, A., Shazeer, N., Parmar, N., Uszkoreit, J., Jones, L., Gomez, A. N., ... & Polosukhin, I. (2017). Attention is all you need. Advances in neural information processing systems, 30.
[5] Ramesh, A., Kirsch, L., van Steenkiste, S., & Schmidhuber, J. (2022). Exploring through random curiosity with general value functions. Advances in Neural Information Processing Systems, 35, 18733-18748.

**Limitations:**

The main limitation is the limited understanding of the generalizability of the results to problem domains, even sparse reward ones, that are different than PixMC-Sparse, which most are.

The paper does address that MIMEx may hurt performance when rewards aren't sparse, but it could provide more limitations on absence of additional memory, compute expense, and wall-clock time needed to add the MIMEx intrinsic reward generation model, which has multiple, multiple-layer transformers and more.

---

> ### Author Rebuttal · Authors · 2023-08-09
>
> Thank you for the feedback. We addressed your main concern on the lack of diverse domains in our empirical study through experiments on two additional discrete-action environments. Our approach performs competitively against baselines, demonstrating its generalizability beyond continuous control tasks. Regarding your concern on the novelty, we would like to highlight that our main contribution is a novel framework that draws connections between masked autoencoding, pseudo-likelihood, and intrinsic rewards, as well as unifies many existing intrinsic bonus approaches. While each component of MIMEx has been explored in the past, the combination of them is what enables superior results. To the best of our knowledge, we are the first to explore using a random masked prediction objective for intrinsic bonus in the context of RL exploration.
>
> **evaluate on more diverse types of sparse rewards e.g. discrete instead of continuous**
>
> We agree with this point, and would like to clarify that our experimental results have already covered (1) tasks with a mix of discrete and continuous rewards and (2) tasks with only discrete rewards. Examples of the former: Pick, quadruped_run; examples of the latter: Reach, cartpole_swingup_sparse. We will add a table summarizing the detailed reward terms and indicating each reward term’s type (discrete/continuous) in our revised manuscript.
>
> **more diverse tasks and environments**
>
> Thank you for the suggestion. To check if MIMEx generalizes to other domains, we ran additional experiments to compare MIMEx with ICM and RND on the ALE *PRIVATE EYE* and *VENTURE* environments. Below, we report all results with 95% confidence intervals, over 5 random seeds on each method (ICM, RND, MIMEx).
>
> [*PRIVATE EYE*]
>
> | mean episodic return\env steps | 10M | 25M | 50M | 100M |200M|
> |----------------------------|-----|-------|-------|---|---|
> | RND                        | -164.8± 279.0  | -188.4 ± 281.9   |  -160 ± 413.9   | 4105.8  ± 112.0  | 6036.4  ±7010.1  |
> | ICM                        | -31.4 ±103.7 | 2334.8 ±2095.9 | 7864.6 ±4605.5 | 11963.8 ± 746.1 | 23377.6  ± 9720.1  |
> | MIMEx                      | -529  ± 416.4 | 104.6  ± 658.8  | 2344.8  ±2776.8  | 7859.6  ± 6866.5 | 7717.2  ± 6239.4 |
>
> [*VENTURE*]
>
> | mean episodic return\env steps | 10M  | 25M | 50M | 100M  |
> |----------------------------|------|------|------|---|
> | RND                        | 580 ± 227  | 1080 ± 144 | 980 ± 73 | 1660 ± 159 |
> | ICM                        | 380 ± 243 | 1240 ± 202 | 1400 ± 310 | 1560 ± 133  |
> | MIMEx                      | 460  ± 320  | 1080 ± 114 | 1040 ± 100  | 1660  ± 48 |
>
> Our results for ICM and RND are similar to those reported in prior works. For MIMEx, we only tuned the exploration beta parameter; without extensive hyperparameter tuning, MIMEx still performs comparably with ICM/RND. We hope these results improve confidence on the generalizability of MIMEx to other domains.
> We will include the updated results and additional code in our revised work.
>
> **MIMEx as a "lightweight" module**
>
> We used the term “lightweight” to imply programmatic simplicity rather than low memory consumption or wall-clock time. We will remove this sentence in the next revision to avoid confusion.
>
> **later exploration methods not used as baselines**
>
> Thank you for the references; we find [2, 3] relevant and will include them in our revised paper. Still, we find our method and baselines more general in formulation, thus more amenable to fair comparison. The strength of MIMEx is that it is a simple module that generalizes a number of prior approaches that are still commonly used in practice, while being more flexible than these approaches. We therefore compare MIMEx against ICM and RND, two existing works that are general in formulation and strong in performance, to obtain performance difference comes from differences in algorithmic formulation rather than implementation details. An interesting direction of future work would be to use other ideas in intrinsic motivation together with our method, such as [2,3].
>
> **better justify the baselines**
>
> We will update the writing to elaborate on why we chose the baselines (as discussed above).
>
> **potential applications beyond the current scope**
>
> We are particularly excited about applying MIMEx to real-world sensorimotor learning; many challenging robotic tasks are difficult to be specified with a dense reward function (e.g. screwing a water bottle cap where un-grasping and re-grasping are needed). We hope MIMEx’s promising results in simulation could transfer to real-world RL. We will add these discussions in the next revision.
>
> **Questions**
>
> 1. $|X|$ is actually always finite here since we define $X$ as categorical variables. In this case, the masking distribution is a discrete uniform distribution and the equality holds.
>
> 2. We use transformer blocks as in [4], with the same order and properties.
>
> 3. We chose these baselines because of their effectiveness and generality across a wide range of domains and tasks.
>
> 4. We include this sample curriculum to illustrate how PixMC-Sparse could be easily modified to avoid saturation as exploration algorithms get more sophisticated. We provide one example of such evaluation in Figure 2 (on *KukaPick*).
>
> 5. [5] derives intrinsic rewards through predicting temporally extended general value functions, though being substantially different in terms of the environment it evaluates on (MiniGrid) and the focus (partial observability). We will include this reference in our next revision.
>
> 6. Thank you for the suggestion. We will do so in our revised version.
>
> 7. In the context of offline RL, online exploration is not possible since offline RL only uses previously collected data without additional online data collection. It is therefore unclear how intrinsic bonus can be applied in that setting.
>
> **Limitations**
>
> We included additional results (see above) and runtime/memory analysis (see global PDF) to address these limitations.

---

### Official Review · Reviewer_kmMp · 2023-07-07

**Soundness:** 3 good
**Presentation:** 2 fair
**Contribution:** 3 good
**Rating:** 5
**Confidence:** 3

**Summary:**

This work proposed a general framework for deriving intrinsic rewards called Masked Input Modeling for Exploration (MIMEx). This method starts from the observation that existing intrinsic reward approaches are special cases of conditional prediction, where the estimation of novelty can be seen as pseudo-likelihood estimation with different mask distributions. From this perspective, MIMEx derives intrinsic reward based on masked prediction on input sequences, which naturally lends itself to controlling the difficulty of the underline conditional prediction task. Empirically results on eight tasks from PixMC-Sparse and six tasks from DeepMind Control Suite demonstrate that MIMEx outperforms other baselines regarding sample efficiency.

**Strengths:**

1. This work proposed a general framework for deriving intrinsic rewards, which can be applied to various hard-exploration tasks.
2. The most interesting part of MIMEx is that it enables extremely flexible control over the difficulty of conditional prediction for deriving intrinsic rewards. Section 5.4 provides comprehensive studies to understand how varying mask distribution affects the performance of MIMEx.
3. Extensive ablation studies have been provided to understand why MIMEx works better than other approaches. The comparisons are provided among diverse directions, including trajectory-level exploration, variance reduction, and model scalability.


**Weaknesses:**

1. The smoothness of writing can be improved, e.g., section 3.1 is not closely related to other parts of MIMEx. The motivation of using masked prediction loss as the intrinsic reward is not clear enough.
2. Line 81 cited a retracted paper.


**Questions:**

1. What motivates using masked prediction loss as the intrinsic reward?
2. When reading the introduction, I have the following questions: how masked language modeling is connected to pseudo-likelihood? Why approaches that estimate novelty can be viewed as modeling different conditional prediction problems or masked prediction problems with different mask distributions?

**Limitations:**

1. One limitation of MIMEx is that it can potentially hurt performance on easier tasks that do not require exploration, as in the case of exploration bonus approaches.
2. Stronger bias from other sources, like the expert demonstrations, is needed to improve the exploration ability of MIMEx further.


------------
After rebuttal
---------------
I would like to thank the authors for addressing part of my concerns. I agree to increase my score slightly.

---

> ### Author Rebuttal · Authors · 2023-08-09
>
> Thank you for the feedback. We addressed individual comments and questions below, in particular your main concern on the validity of our method due to the citation of a retracted paper. We clarified that the key principle our method depends on is independent of the error in the retracted paper. Please let us know if there are any remaining issues that prevent our paper from getting a better score.
>
> **”The smoothness of writing can be improved.”**
>
> In section 3.1, we describe details of MIMEx’s formulation and architecture in the form of a sequence autoencoder; we see that the section title could be potentially confusing, and will update it in our revised manuscript to “Sequence-Level Masked Autoencoders” to make for a better summary of the section and a smoother transition from the former paragraph. We will also update the writing to strengthen the motivation of using masked prediction loss as the intrinsic reward (as we will elaborate under “Questions”) and improve the overall smoothness in our next revision.
> Please let us know if you find any other specific places where writing could be improved, and we will look into improving them accordingly.
>
> **”Line 81 cited a retracted paper.”**
>
> As we noted on line 81 (with a footnote that points to Appendix A.1) in our submission, while [40] was later retracted due to an error, we are aware of the error and our claims remain valid despite the error in cited work. In [40], the authors mistakenly viewed BERT [9] as a Markov Random Field (MRF). While the original goal of [40] was to derive a procedure for sampling from masked language models (MLMs) by viewing them as MRFs, the work also inspired the use of MLM prediction error as a way for scoring sequences when decoding from a language model. The way we propose to use MLMs is similar to the latter, i.e. as a proxy metric for scoring sequence-level predictions of trajectory information. In other words, we do not formally treat the MLM in MIMEx as an MRF, and we do not attempt to obtain conditional distributions from which one generates samples. We also note that a correct energy-based view was later proposed in [13], which does not change the argument that we put forth either. While sampling from an energy-based model is expensive, we only seek to obtain a useful stochastic estimate of the energy function for the purpose of scoring. We will clarify the relationship between our method and [40] further in our next revision of the paper.
>
> **Questions**
>
> **1.** As we discussed in Section 3, the use of masked prediction loss is primarily motivated by its flexibility and effectiveness (as demonstrated by empirical results). Masked prediction can be applied to input sequences with arbitrary length and at arbitrary mask distribution; such a framework naturally lends itself to greater control over the difficulty of the underlying conditional prediction problem. By setting up conditional prediction problems on trajectories, we can obtain intrinsic rewards that consider transition dynamics across longer time horizons and extract richer exploration signals. We can also easily tune the difficulty of the prediction problem, by varying both the input length and the amount of conditioning context given a fixed input length. (Meanwhile, existing approaches framed as conditional prediction often consider one-step future prediction problems, which can saturate early as a useful exploratory signal; longer time-horizon prediction problems capture more complex behavior, but they can suffer from high variance.)
>
> Additionally, we are motivated by the generality and scalability of the masked autoencoding objective. Masked autoencoding relies on less domain knowledge compared to methods like contrastive learning, and has proven success across many different input modalities.  We can also leverage standard architectures such as those used in masked language modeling and masked image modeling, for which the scalability and stability have been tested.
>
>
> **2.** Masked language modeling is connected to pseudo-likelihood through the estimation of conditional distributions. Pseudo-likelihood is a way of approximating likelihood by modeling the conditional distributions of variables given all other variables. Masked language modeling, such as the masked autoencoding objective in models like BERT, can be seen as a form of stochastic maximum pseudo-likelihood estimation, as we illustrated in section 2.3.
>
> In masked language modeling, a model is trained to predict masked or missing tokens in a sequence. The model approximates the underlying joint distribution among variables by modeling the conditional distributions of the masked tokens given the rest of the sequence. By optimizing the model to predict the masked tokens, it effectively estimates the likelihood of the observed sequence. Approaches that estimate novelty can therefore be viewed as modeling different conditional prediction problems or masked prediction problems with different mask distributions; Table 1 provides several such examples. By applying different mask distributions, which determine the pattern of masking in the input data, various aspects of novelty can be captured. For example, masking a subset of state variables can measure novelty in states, while masking current or next-step features can estimate novelty in state transitions. Each approach represents a different way of modeling the conditional distributions and approximating the likelihood of the masked components.
>
> **Limitations**
>
> Thank you for summarizing the limitations that we mentioned in Section 7 of the manuscript. We have been working on improving our exploration framework such that it can be generalizable to scenarios where exploration is not required and draw on stronger bias from sources like expert demonstrations, and will report the method and results in our follow-up works.

---

> > ### Comment · Reviewer_kmMp · 2023-08-17
> > **Reply to author**
> >
> > Thank you for addressing part of my concerns. I agree to increase my score slightly.

---

### Author Rebuttal · Authors · 2023-08-09

We thank all the reviewers for their valuable feedback. Below, we respond to each reviewer individually.

In the PDF attached below, we include additional results on wall-clock time/GPU memory usage (reviewer mFvm, reviewer SVND) and mask distribution ablation studies (reviewer if59).

---

### Decision · Program_Chairs · 2023-09-21

**Decision:**

Accept (poster)

**Comment:**

While there is a bit of variance in the reviews, everyone recommended the paper for acceptance and the authors did a good job in rebuttal pushing the ratings up. I also agree with this assessment. There are clear room for improvement as mentioned by reviewers but the paper's contribution passes the bar to get accepted.